# Analysis of the Interactome of the *Toxoplasma gondii* Tgj1 HSP40 Chaperone

**DOI:** 10.3390/proteomes11010009

**Published:** 2023-03-01

**Authors:** Jonathan Munera López, Andrés Mariano Alonso, Maria Julia Figueras, Ana María Saldarriaga Cartagena, Miryam A. Hortua Triana, Luis Diambra, Laura Vanagas, Bin Deng, Silvia N. J. Moreno, Sergio Oscar Angel

**Affiliations:** 1Laboratorio de Parasitología Molecular, INTECH, CONICET-UNSAM, Av. Intendente Marino Km. 8.2, C.C 164, Chascomús 7130, Argentina; 2Center for Tropical and Emerging Global Diseases, University of Georgia, Athens, GA 30602, USA; 3Centro Regional de Estudios Genómicos, Universidad Nacional de La Plata, La Plata 1900, Argentina; 4Department of Biology and VBRN, University of Vermont, Burlington, VT 05405, USA; 5Department of Cellular Biology, University of Georgia, Athens, GA 30602, USA

**Keywords:** *Toxoplasma*, Hsp40, protein–protein interaction, Hsp70/Hsp90 cycle

## Abstract

*Toxoplasma gondii* is an obligate intracellular apicomplexan that causes toxoplasmosis in humans and animals. Central to its dissemination and pathogenicity is the ability to rapidly divide in the tachyzoite stage and infect any type of nucleated cell. Adaptation to different cell contexts requires high plasticity in which heat shock proteins (Hsps) could play a fundamental role. Tgj1 is a type I Hsp40 of *T. gondii*, an ortholog of the DNAJA1 group, which is essential during the tachyzoite lytic cycle. Tgj1 consists of a J-domain, ZFD, and DNAJ_C domains with a CRQQ C-terminal motif, which is usually prone to lipidation. Tgj1 presented a mostly cytosolic subcellular localization overlapping partially with endoplasmic reticulum. Protein–protein Interaction (PPI) analysis showed that Tgj1 could be implicated in various biological pathways, mainly translation, protein folding, energy metabolism, membrane transport and protein translocation, invasion/pathogenesis, cell signaling, chromatin and transcription regulation, and cell redox homeostasis among others. The combination of Tgj1 and Hsp90 PPIs retrieved only 70 interactors linked to the Tgj1-Hsp90 axis, suggesting that Tgj1 would present specific functions in addition to those of the Hsp70/Hsp90 cycle, standing out invasion/pathogenesis, cell shape motility, and energy pathway. Within the Hsp70/Hsp90 cycle, translation-associated pathways, cell redox homeostasis, and protein folding were highly enriched in the Tgj1-Hsp90 axis. In conclusion, Tgj1 would interact with a wide range of proteins from different biological pathways, which could suggest a relevant role in them.

## 1. Introduction

*Toxoplasma gondii* is an obligate intracellular protozoan parasite that belongs to the phylum Apicomplexa, which includes parasites characterized by a polarized cellular structure and a particular arrangement of its cytoskeleton and apical organelles [1]. It can infect almost any nucleated cell in warm-blooded animals, including humans. *T. gondii* is the causative agent of toxoplasmosis [1], a worldwide distributed parasitic zoonosis. It is estimated that up to one-third of the world’s human population has been exposed to the parasite and that between five hundred million and one billion humans are chronically infected [2]. However, the clinical disease is mostly restricted to two risk groups: immunocompromised patients and individuals who acquired the infection by vertical transmission (congenital toxoplasmosis) [2,3]. The parasite has a complex life cycle, alternating between definitive hosts (any member of the cat family) and intermediate hosts (likely all warm-blooded animals, including humans). Inside intermediate hosts, *T. gondii* has two stages of asexual development: the rapidly growing tachyzoite, responsible for acute illness, and the slowly dividing bradyzoite, responsible for chronic and asymptomatic infection [4]. The tachyzoite can invade any nucleated cell, where it replicates rapidly within a parasitophorous vacuole [5], increasing in number exponentially until cell lysis. They egress from the host cell to infect other cells, repeating the lytic cycle.

The infectious process relies on the ability of *T. gondii* to recognize different cell types (intestinal epithelium, macrophages, neurons, muscle cells, etc.) and adapt to a variety of environmental conditions. In each cellular context, the tachyzoite must carry out its biological processes such as DNA replication, transcription, translation, invasion, and egress, among others. In this sense, heat shock proteins (Hsps) would be expected to be essential for a successful infection. The chaperones Hsp90, Hsp70, and some Hsp40s were shown to be essential for the lytic cycle [6].

Hsps are key molecules in living organisms that regulate protein folding, and prevent the aggregation of misfolded proteins, among other functions. Hsp40, Hsp70, and Hsp90 facilitate diverse cellular processes such as protein folding, protein translocation across membranes, signal transduction, DNA replication, DNA damage response, and protein degradation [7,8,9].

Among the Hsps, the Hsp40 family (also called J-proteins) is classified into four types according to their domain organization [10,11]. Type I (DNAJA1) J-proteins can act as molecular chaperones, binding and delivering non-native proteins to the Hsp70 protein. Among them, *Saccharomyces cerevisiae* Ydj1 and human Hjd-2 were the most studied. Ydj1 was found primarily in the cytosol, concentrated in a perinuclear ring, and co-fractionated with nuclei and microsomes [12]. In addition to a J-domain, Ydj1 and Hdj2 present a CaaX C-terminal motif which is specific for farnesylation at a cysteine residue [12,13,14,15]. Farnesylation facilitates the attachment of Ydj1 to the endoplasmic reticulum (ER) and mitochondrial membranes. This process assists in the transport of mitochondrial and type II secretory/membrane proteins from the cytoplasm to mitochondria and the ER, respectively [16,17,18]. By using a synthetic lethal approach, Gillies et al., [19] observed that Ydj1 has a generalist role together with Hsp90.

*T. gondii* Hsp40 represents a large and diverse family of 36 identified members, one of them identified as a putative Ydj1-like Hsp40 [20]. Here, we present the characterization of *T. gondii* Ydj1-like Hsp40, renamed Tgj1. The subcellular localization of Tgj1 was analyzed with an anti-Tgj1 antibody produced in mice using the recombinant protein. The Tgj1 and *T. gondii* interactome under normal growth conditions was performed by co-immunoprecipitation (co-IP) and mass spectrometry analysis. Additionally, Tgj1 and *T. gondii* Hsp90 PPIs were combined to infer the Tgj1-(Hsp70)/Hsp90 axis chaperone network. Finally, we discuss the role of Tgj1 in different biological processes.

## 2. Materials and Methods

### 2.1. Parasite Culture

RH∆HXGPRT [21] and transfected lines of *T. gondii* were cultured in standard tachyzoite conditions in vitro in hTERT (BD Biosciences) fibroblast monolayers [22]. Monolayers were infected with tachyzoites and incubated in Dulbecco’s modified Eagle medium (DMEM, GIBCO) supplemented with 1% fetal bovine serum, penicillin (100 UI/mL; GIBCO), and streptomycin (100µg/mL; GIBCO) at 37 °C and 5% CO_2_. 

### 2.2. Alignment, Phylogenetic, and Domain Analysis

Sequences were aligned by Clustal W (Bioedit Sequence Alignment Editor 7.0.5.3 version). Pair Distance was analyzed by MegAlign software (Dnastar). The phylogenetic tree was constructed with MEGA-X (Molecular Evolutionary Genetics Analysis Software, version X). Internal support was measured using 1000 replicates of the heuristic search bootstrap option. The evolutionary history was inferred using the UPGMA method. The tree was condensed (cutoff value: 50%). Domains were retrieved by Motif Scan “https://myhits.sib.swiss/cgi-bin/motif_scan (accessed on 11 october 2022)”.

### 2.3. Expression and Purification of Recombinant Tgj1 

Total RNA from tachyzoites (RH strain) was extracted using TRIzol^®^ Reagent (Life Technologies) according to the manufacturer’s instructions. Between 1 and 5 µg of RNA were used to produce the corresponding cDNA through reverse transcription reactions using 100 U of MMLV reverse transcriptase (Life Technologies), and oligo-dT as a primer for mRNA. The following primers were used for the PCR reactions: Tgj1 (TGME49_311240) sense (Fw): 5′-GGATCCATGTATTTTGGCAGCTTC-3′ and Tgj1 antisense (Rv): 5′-AAGCTTATTGGTGTATACGCGGTCTTCT-3′ (which recognizes a region on the 3′UTR sequence). KpnI and HindIII sites were added (underlined sequence), respectively. The amplified product (Tgj1 open reading frame [ORF]) was cloned in the cloning vector pGEM-T easy (Promega) and sequenced (Macrogen Corp., Seoul, Republic of Korea).

Tgj1 was expressed and purified as recombinant protein (rTgj1) as previously described [20]. Briefly, Tgj1 ORF was subcloned into the prokaryotic expression vector pRSET-B (Life Technologies, Argentina) in frame with a sequence that encodes for 6 N-terminal histidines (6 His-tag). *Escherichia coli* BL21 (DE3) pLys (Novagen) bacteria were transformed with the expression plasmid. Cultures were grown to OD600 = 0.4–0.6 before protein expression induction by the addition of 2 mM isopropyl thio-β-D-galactoside (IPTG). After overnight incubation at 37 °C, the cells were harvested and purified using a commercial His-Trap 5 mL column (GE Healthcare) according to the manufacturer’s instructions (under non-native conditions).

### 2.4. Primary Antibody Sources

The source of rabbit anti-TgHsp90 antibody was already described [23]. Guinea pig anti-TgSERCA (*T. gondii* sarcoplasmic-endoplasmic reticulum calcium ATPase antibody) was already described [24]. To obtain anti-rTgj1 antibodies, mice were immunized with rTgj1 (100 μg per dose). Three boosters of the antigen, emulsified with incomplete Freund’s adjuvant at intervals of 2 weeks, were performed following a primary immunization performed with complete Freund’s adjuvant. Samples of pre-immune serum were collected from each animal before antigenic stimulation [20]. The animal procedure was approved by the Animal Welfare Committee of the Universidad de General San Martín (C.I.C.U.A.E., IIB-UNSAM, 09/2016). Mice had access to food and water ad libitum and were maintained at 21–22 °C, 12:12 h light-dark photocycle.

### 2.5. Immunoblot Analysis

Approximately 0.5–1 × 10^9^ tachyzoites purified from the host cell material by passage through a 3 µm pore size filter (Amersham Hybond, GE Healthcare, Ciudad Autónoma de Buenos Aires, Argentina) were used to prepare parasite lysate. Proteins were analyzed by sodium dodecyl sulfate-polyacrylamide gel electrophoresis (SDS-PAGE) on 10% acrylamide gels in the Mini-Protean system (Bio-Rad). After electrophoresis, the proteins were transferred to PVDF (polyvinylidene difluoride) (Millipore Corp., Middlesex County, MA, USA) for immunoblot analysis. Electrotransfer conditions were 1h at 100V in cold transfer buffer (25 mM Tris, 192 mM glycine, 10% methanol) with a Bio-Ice cooling unit. Non-specific binding sites were blocked with 5% non-fat-dried milk in Phosphate-buffered saline (PBS) containing 0.1% Tween-20 (PBS-T) and the membranes were then incubated for 1h at room temperature or overnight at 4 °C with the corresponding primary polyclonal antibodies diluted 1:500. After that, membranes were incubated with alkaline phosphatase-conjugated anti-rabbit or anti-mouse secondary antibodies, diluted at 1:10,000 (Santa Cruz Biotechnology, Dallas, TX, USA). Immunoreactive protein bands were visualized by the NBT-BCIP method (Sigma-Aldrich^TM^, Ciudad Autónoma de Buenos Aires, Argentina). 

### 2.6. Co-Localization Analysis

Co-localization analyses have been described before [23]. Briefly, tachyzoites were grown in vitro in hTERT. In the case of Hsp90/Tgj1 co-localization, intracellular tachyzoites were fixed with 3% paraformaldehyde for 15 min, followed by permeabilization with 0.25% Triton X-100 for 10 min and blocking with 3% bovine serum albumin (BSA) at room temperature for 30 min. Primary antibodies were mouse anti-Tgj1 at 1:200 dilution and for co-localization rabbit anti-TgHsp90 at 1:200 dilution (cytosolic marker). After incubation with primary antibodies, cells were washed three times with PBS 1× and then incubated with the corresponding secondary antibodies (1:4000) Alexa fluor goat anti-rabbit 594 or Alexa fluor goat anti-mouse 488 (Invitrogen). Coverslips were washed three times and mounted in Fluoromount G^®^ (SoutherBiotech, Birmingham, AL, USA) containing 2 μg/mL 4′,6-diamino-2-phenylindole (DAPI). In the case of TgSERCA/Tgj1 co-localization, extracellular parasites were harvested, filtered, and washed once with Buffer A with glucose (BAG, 116 mM NaCl, 5.4 mM KCl, 0.8 mM MgSO4, 50 mM HEPES, and 5.5 mM dextrose). After that, 50 µL aliquots containing 2 × 10^4^ parasites were overlaid on coverslips previously treated with 1 mg/mL poly-L-Lysine. Parasites were then fixed with 3% paraformaldehyde for 15 min, followed by permeabilization with 0.25% Triton X-100 for 10 min, and blocking with 3% BSA at room temperature for 30 min. Primary antibodies were mouse anti-TgJ1 at 1:500 dilution and for colocalization guinea pig anti-TgSERCA-ATPase at 1:500 dilution (ER marker). Secondary antibodies were goat anti-mouse 488 and goat anti-guinea pig Alexa-Fluor596. Slides were mounted with Fluoromount G^®^ (SoutherBiotech, Birmingham, AL, USA) containing 2 μg/mL DAPI. Images were taken with an Olympus IX-71 inverted fluorescence microscope with a Photometric CoolSnapHQ CCD camera driven by DeltaVision software (Applied Precision, Seattle, WA, USA, magnification 100×, NA 1.40 at 24 °C). Green, red, and blue fluorescence were recorded separately and the images were colored by Image-Pro Plus version 5.1.0.20 and analyzed and merged using Adobe Photoshop. Parallel reactions of an unrelated antibody (anti-HA antibody) were used for evaluating the unspecific background.

### 2.7. Co-Immunoprecipitation (co-IP) Analysis

Co-IP assays were performed as follows: 0.5–1 × 10^9^ tachyzoites were resuspended in 1–2 mL of ice-cold receptor buffer (RB: 50 mMTris–HCl pH 7.5, 50 mM NaCl, 1 mM EDTA, 1 mM DTT, 10% glycerol, 10 mM Na-molybdate, protease inhibitor cocktail: 1 µg/mL each of leupeptin, aprotinin, pepstatin A, and 1 mM PMSF) and immediately sonicated at 30% amplitude in cycles of 15 s ON and 10 s OFF. Then, the samples were centrifuged for 15 min at 10,000× *g*, 4 °C, and the supernatant was recovered as the protein extract, from which 10% was used as INPUT. Primary anti-Tgj1 or -*T. gondii* Hsp90 (25 μL) was added and incubated overnight at 4 °C with rocking. A preimmune serum sample was also used as a control. Then, 50 µL of 100 mg/mL Protein A-Sepharose (Santa Cruz Biotechnologies, Texas, USA) previously equilibrated in RB was added and the incubation continued for 60 min. The beads were spun, and the supernatant was carefully discarded. The beads were washed 3× with 1 mL buffer RB with 0.1%Triton X-100. To reduce the background, the beads were transferred to a fresh tube after the first wash. The final wash was performed with RB buffer without Triton X-100 and the supernatant was carefully removed. The beads were finally resuspended in 30 µL 2.5 × Laemmli sample buffer, boiled for 3 min, spun for 5 min, and loaded on SDS-PAGE. Shared bands between co-IP with preimmune and co-IP with anti-Tgj1 or anti-*T. gondii* Hsp90 were discarded, mainly immunoglobulins. 

### 2.8. Separation and Digestion of Proteins

Protein samples were separated by 1D SDS-PAGE in the Mini-Protean system (Bio-Rad Laboratories, Inc., Ciudad Autónoma de Buenos Aires, Argentina) with a thickness of 1.5 mm. The resultant gel was stained with colloidal *Coomassie Brilliant Blue* G-250 (Sigma-Aldrich^TM^, Ciudad Autónoma de Buenos Aires, Argentina). Protein gels were fixed by soaking for 3 hr in 30% methanol, 2% phosphoric acid. Then, the gel was washed with deionized water 3 times, 5 min each. After removing the deionized water, the staining solution (0.5 g/L Coomasie blue brilliant R250, 18% methanol, 17% (NH_4_)_2_SO_4_, 2% phosphoric acid) was added until covering the gel. It was stained for 1 h in gentle shaking. Then, the gel was rinsed in deionized water 3 times for 5 min each. The background staining was removed with 30% methanol. The gel was stored in deionized water until use. Each lane of the gel was cut into individual slices. Each band was then cut into 1 mm^3^ cube and further treated with three washes of 50 mM NH_4_HCO_3_ in 50% CH_3_CN with 10 min incubations. Each group of gel cubes was then dehydrated in CH_3_CN for 10 min and dried in a Speed Vac. Protein samples were reduced by dithiothreitol (DTT) and alkylated by iodoacetamide [25]. A solution of 10 ng/µL trypsin in 50 mM NH_4_HCO_3_ was used to re-swell the gel pieces completely at 4 °C for 30 min, followed by a 37 °C digestion overnight. A small amount of 10% formic acid was then added to stop the digestion. The sample was then centrifuged at 2800× *g*, and the supernatant was collected for LC-MS/MS. The co-IP analysis was done by two independent sources of RH tachyzoites lysates for Tgj1 and Hsp90. In the case of Tgj1, two different co-IPs were made per lysate (Appendix A).

### 2.9. LC-MS/MS Analysis

A fused silica microcapillary LC column (15 cm long × 75-µm inside diameter) packed with Halo C18 reversed-phase resin (2.7 µm particle size, 90 nm pore size, MichromBioresources.) was used with EASY-nLC 1200 system (Thermo Fisher, Waltham, MA, USA). The nanospray ESI was fitted onto the Thermo Q-Exactive plus mass spectrometer (Thermo Electron, San Jose, CA, USA) that was operated in a Higher-energy C-trap dissociation mode to obtain both MS and tandem MS (MS/MS) spectra. Two µL of tryptic peptide samples were loaded onto the microcapillary column and separated by applying a gradient of 3–40% acetonitrile in 0.1% formic acid at a flow rate of 300 nL/min for 80 min. Mass spectrometry data were acquired in a data-dependent top-10 acquisition mode, which uses a full MS scan from *m*/*z* 350–1700 at 70,000 resolution (automatic gain control [AGC] target, 1 × 10^6^; maximum ion time [max IT], 100 ms; profile mode). Resolution for dd-MS2 spectra was set to 17,500 (AGC target: 1 × 10^5^) with a maximum ion injection time of 50 ms. The normalized collision energy was 27 eV. 

### 2.10. Protein Identification

Obtained MS spectra were searched against the *T. gondii* ToxoDB-42 _TgondiiME49 protein database using Proteome Discoverer 2.2 (Thermo Electron, San Jose, CA, USA). The search parameters permitted a 10 ppm peptide MS tolerance and a 0.02 Da MS/MS tolerance. Carboxymethylation of cysteines was set as a fixed modification and oxidation of methionine as a dynamic modification. Up to two missed tryptic peptide cleavages were considered. The proteins for which the False Discovery Rate was less than 1% at the peptide level were included in the following analysis. The raw data of all mass spectra had been submitted to the Pride (http://www.ebi.ac.uk/pride/archive; Project Name: Toxoplasma gondii Tgj1 Hsp40 protein interaction LC-MS/MS; Project accession: PXD024656).

### 2.11. Bioinformatics

GO terms, LOPIT [26], and metabolic pathways were obtained from www.toxodb.org (accessed on 11 october 2022). Protein–protein Interaction maps were performed by STRING software (https://string-db.org/, accessed on 11 october 2022). Word cloud plots were constructed by a Wolfram script; the size of the words is proportional to the frequency of the biological term.

## 3. Results 

### 3.1. Sequence Analysis of T. gondii j1 (Tgj1), an Essential Hsp40

To confirm that TGME49_311240 (here renamed as Tgj1) is a Ydj1/Hdj2 homolog, a phylogenetic tree was built and included other *S. cerevisiae* type I Hsp40 sequences, such as Apj1 and Xdj1, both closely related to Ydj1 but functionally divergent [27] and *Plasmodium* PF3D7_1437900 (PF14_0359) cytosolic type I Hsp40 [28], renamed ERdj3 [29]. Tgj1 and Erdj3 sequences clustered with Ydj1, and Hdj2, suggesting that both are Ydj1/Hdj2 homologs (Figure 1A). As expected, the alignment of the deduced amino acid sequence of Tgj1 with the amino acid sequences of ERdj3, Ydj1, and Hdj2 shows a high identity (Figure 1B). The Tgj1 sequence compared with Hdj2 and Ydj1 is 46.4%, and 45.2% identical, respectively; while Ydj1 and Hdj2 have an identity of 48.7% (Appendix A). A similar comparison of Tgj1 and Apj1 and Xdj1 amino acid sequences showed 31.7% and 30.0% identity. In addition, other *T. gondii* type I Hsp40s (TGME49_207760 and TGME49_258390) [20] presented similarities below 35.6 (Appendix A). Based on this analysis, the protein encoded by TGME49_311240 (Tgj1) gene was considered as a Ydj1/Hdj2 homolog. 

Tgj1 encodes a protein of 426 amino acids with a predicted molecular mass of 48 kDa and the characteristic structural domain organization such as Ydj1/Hdj2 (Figure 1C): an N-terminal J domain with histidine-proline-aspartic acid (HPD) tripeptide motif, a central ZFD with 4 “CXXCXG” zinc finger motifs, a DNAJ_C domain. In addition, it includes most of the C–terminal hydrophobic residues relevant for homodimerization as observed in yeast [30] as well as the conserved last four CRQQ residues motif, which resembles the CaaX motif (where “C” is cysteine, “a” is usually an aliphatic amino acid and “X” is any amino acid). The CaaX motif is present in proteins that are usually farnesylated, among them ERdj3, Ydj1, and Hdj2 [14,16,29,31]. Tgj1 (TGGT1_311240) shows a phenotype score of −3.91 (a negative score correlates with fitness conferring) [6], suggesting that is essential for the lytic cycle.

### 3.2. Subcellular Localization of Tgj1

Subcellular co-localization of Tgj1 and Hsp90 was analyzed by using a murine anti-Tgj1 antibody (Figure 2A). Both Hsp90 and Tgj1 were cytosolic (Figure 2B). Since farnesylated Ydj1 and Hdj2 also showed ER membrane localization [16,17], co-localization with ER was also studied using anti-TgSERCA antibody as ER marker [24]. Tgj1 labeling was mostly cytosolic but with regions co-localizing with the ER-resident protein SERCA (Figure 2C), suggesting that a Tgj1 fraction could be associated with ER. 

### 3.3. Identification of Putative Tgj1 Interacting Proteins in T. gondii

To determine the protein–protein interaction (PPI) pattern of Tgj1, a co-IP assay was done using the anti-Tgj1 antibody followed by SDS-PAGE and mass spectrometry (MS) analysis of sliced bands. Only proteins showing two or more unique peptides by MS, and detected in two independent co-IPs, were further considered. We identified 336 putative interacting proteins for Tgj1 (Appendix A). As expected, proteins from different subcellular localizations, biological processes, and metabolic (KEGG) pathways were pulled down (Figure 3, Appendix A). 

Recently, a subcellular localization analysis by LOPIT was incorporated into the ToxoDB [26]. LOPIT is an experimental approach that also allowed the incorporation of subcellular location data for more genes than those determined by gene ontology (GO) terms. Using this tool, the subcellular location of geneIDs detected in the Tgj1 co-IP/MS was predicted (Appendix A). The number of genes detected by LOPIT in each cell compartment was analyzed, and then the percentage of enrichment of the interactors was calculated (Appendix A, tab %Tgj1). An enrichment for interactors of different subcellular locations was observed, mainly with those linked to the cytoskeleton, ribosomes, and cytosol, but also with proteins associated with membranes such as plasma membrane, ER/Golgi, inner membrane complex (IMC), and with organelles such as mitochondria, micronemes, and rhoptries (Figure 3A). 

A total of 151 geneIDs (45%) showed GO terms for Biological Process (BP) and only 58 geneIDs (17.2%) for a metabolic process pathway (Appendix A). Tgj1 seems to participate in several biological processes such as amides, oxoacids, peptides, nitrogenous compounds, and carboxylic acid metabolism (Figure 3B). Amides are common in nature and are found in substances such as amino acids, proteins, DNA and RNA, hormones, and vitamins, and can also participate in energy pathways. In concordance, Tgj1 interactors were associated with nitrogen metabolism and energy generation (Figure 3C). Based on their GOs we constructed a PPI network by STRING (Figure 4). Two large PPI networks can be observed: energy metabolism and protein processing (translation and proteolysis). 

### 3.4. Tgj1-Hsp90 Axis Network

Both Hsp40 Ydj1/Hdj2 (Type I) and Hdj1 (type II) can participate in the recognition of the client protein that is then processed by the Hsp70/Hsp90 cycle [32]. We had previously described the interactome for *T. gondii* Hsp90 with 317 interacting proteins [33]. Similarly to what was done with Tgj1, here we performed a co-IP experiment with anti-*T. gondii* Hsp90, obtaining 149 interactors, of which 100 were shared with the previous interactome. We joined both interactomes to generate a list of 366 interactors for *T. gondii* Hsp90. To analyze the PPI network between the different chaperones, we combined the interactors of these two chaperones (Appendix A). Tgj1 and *T. gondii* Hsp90 share 102 interactors, while each of them has a large number of specific interactors (Figure 5A). 

Since many proteins have specific functional roles in *T. gondii*, based on product description, LOPIT subcellular localization information, GO functions, and in some cases GO molecular function and bibliography, we generated an arbitrary analysis that links the GeneIDs with known *T. gondii* biological pathways (Appendix A). Taken all together, we could infer that Tgj1 (Tgj1 + Tgj1-Hsp90) has a pleiotropic role, mainly associated with translation, protein folding and protein turnover, energy and ATP generation produced at mitochondria, but also cell invasion/pathogenesis, motility, cytoskeleton, membrane, and vesicle transport. Pathways linked to cell signaling as well as nuclear RNA processing, transcription, and chromatin regulation, also seem to be significantly represented. 

When we separately analyze the biological pathways of interactors from the Tgj1-Hsp90 intersection, and the interactors of Tgj1 and Hsp90 by subtracting those of Tgj1-Hsp90, we can infer the specific pathways of Tgj1 from those associated with the Tgj1-Hsp90 axis (Figure 5B). We can also observe the interactors associated with Hsp90 not related to the Tgj1-Hsp90 axis, which could be associated with the Hsp70/Hsp90 cycle initiated by another Hsp40 different from Tgj1, or by Hsp90 alone. The biological pathways of the Tgj1-Hsp90 axis were enriched with translation, protein folding, and tRNA aminoacylation/translation (Figure 5B). A more detailed analysis showed that cell redox homeostasis pathway was also enriched in Tgj1-Hsp90 PPI in comparison with Tgj1 specific interactors, whereas energy pathway and cell shape/motility (IMC and myosin proteins) were enriched in Tgj1 specific PPI in comparison with Tgj1-Hsp90 PPI. Similarly, invasion/pathogenesis (microneme, ropthry, and dense granule proteins) also appears to be enriched in Tgj1 although the Tgj1-Hsp90 PPI showed a high number of interactors for this biological pathway. In contrast, in the group of Hsp90 interactors not related to the Tgj1-Hsp90 axis, protein folding, RNA processing, cell signaling, and transcription stand out compared to the other two PPIs. All the data together suggest that each chaperone or complex would have specialized roles in various biological processes.

## 4. Discussion

We have previously reported sequences of 36 putative Hsp40 proteins in *T. gondii*, which is consistent with the fact that Hsp40 (J-domain) is a large family of proteins [20]. In P. falciparum 43-44 members were reported [33]. Although the role of Hsp40 is to stimulate the ATPase activity of Hsp70, it is expected that the different members also have specific functions and client protein targets, including J-domain independent functions [34,35]. In this work, we characterized an Hsp40 DNAJA1-like protein in *T. gondii* (Tgj1), an ortholog of human (Hdj2), and yeast (Ydj1) chaperones.

Tgj1 presents a major cytosolic localization but also overlaps partially with ER as it was observed for Ydj1 [12,16], although the presence of Tgj1 in the nucleus in low amounts cannot be ruled out. In agreement with the localization of Tgj1 by immunofluorescence, the great majority of the interactors detected by LOPIT were cytosolic or membrane-bound, while a minority were nuclear. 

The Tgj1 PPI network included proteins from different subcellular compartments, with most of them being part of essential metabolic pathways and translation machinery. Confirmation of the interactions should include additional experiments. However, a large number of interactors and different biological pathways was expected because the major function of Hsp40 proteins is to bind different client or unfolded proteins and to regulate adenosine triphosphate (ATP)–dependent polypeptide binding by Hsp70 protein [36,37]. The role of Tgj1 in essential biological pathways is also consistent with its predicted essentiality, as shown by its negative phenotype score (Tgj1 TGGT1_311240: −3.91) [6]. It should be noted that the Tgj1 protein could have different PTMs that could have different biological functions. Our antibody was produced from a recombinant protein from *E. coli*, a system that would not incorporate any of the posttranslational modifications to the protein produced, likely expressing only the native version. However, this antibody should pull down all forms of Tgj1, therefore our proteome is of proteoforms, not canonical proteins. Similarly, the putative interactors identified could represent particular proteoforms.

Most of the PPI network is located in the tachyzoite cytosol, mitochondria, ER, and vesicles; while there are a few in the nucleus. They are associated with different pathways that converge in heat shock response, translation and protein metabolism (ribosomes, elongation, and initiation factors, amino acid tRNA synthetase, and protein folding), energy metabolism (Glycolysis, TCA cycle, Pyruvate metabolism), vesicle transport and ER membranes, cell signaling, transcription, and chromatin modulation, among others. Since there are no co-IP interactomes of Ydj1 or Hdj2 it is difficult to establish if Tgj1 presents novel functions in *T. gondii*, except for those interactors associated with invasion/pathogenesis (rhoptry, microneme, and dense granules proteins). An interaction study in heat-shocked yeast identified Hsps (protein folding pathway) and mitochondrial proteins as interactors of Ydj1 [38]. A synthetic lethality study showed that Ydj1 has a generalist role, associated with translation, cell signaling, membrane integrity, and DNA damage [19]. A systematic global interactive network study in *S. cerevisiae* showed that Ydj1 is associated with vesicle transport proteins, cell polarity and morphogenesis, glycolysis, protein folding, mitosis, fatty acid biosynthesis, metabolism, transcription, and chromatin [39].

Ribosomal proteins are common contaminant interactors. However, almost all of the large ribosome subunits were pulled down with Tgj1, indicating that such interaction would be consistent. Brodsky et al. [40] observed that Ydj1 affects the translation of a subset of proteins in yeast. Although they observed that Ydj1 would weakly bind to ribosome subunits, Ydj1 affected the translation process, especially of mRNAs with rare codons, suggesting that it might be required to charge poorly used tRNAs, to fold specific tRNA synthetases, to catalyze the binding of charged tRNAs to the ribosome, among others. Here, we observed that not only Tgj1 interacts with ribosomal subunits, but also with several amino acid tRNA synthetases. This indicates an important association of Tgj1 with the translation machinery. It could also include interaction with the nascent peptide and its maturation in the cytosol and/or ER (see below), as was observed for Ydj1 [41]. Further studies are needed to elucidate the Tgj1 role in the protein translation process.

Despite the partial ER co-localization of Tgj1, the PPI analysis strongly associated Tgj1 with vesicles, ER and Golgi compartments, mitochondria, membrane transport, and membrane translocation. Proteomics studies deposited in ToxoDB also localized Tgj1 in the mitochondrion matrix, cytosol, membranes, and extracellular vesicles [42,43]. Similar subcellular localization was also observed in Ydj1. Interestingly, Ydj1 mutants that cannot be farnesylated have shown defects in the transport of polypeptides across the membrane of the mitochondria and the ER, and in interaction with client proteins of the Hsp70/Hsp90 cycle [16,17]. 

Importantly, the CRQQ motif is not exactly the canonical CaaX, where “C” is cysteine, “a” is typically an aliphatic amino acid, and “X” can be one of several residues. Several reports showed that CaaX motifs are not prenylated, although this entire motif, or at least the cysteine, is essential for the correct role of the protein [44,45,46]. Additional studies are necessary to determine if the CRQQ motif is functional in Tgj1, linked to ER partial localization of Tgj1.

The participation of Tgj1 with the translation machinery and membrane-associated proteins could be related to its role in assisting the translocation of ER proteins. Proteins that are secreted, lysosomal, or integrated into plasma membranes and ER/Golgi must be translocated to the ER co-translationally via signal peptide (SP) or post-translationally associated with Ydj/Hdj2 type chaperones [47]. In both cases, they do so through the Sec61 translocation channel. The Sec62 protein, on the other hand, could facilitate post-translational protein translocation [47]. An association with ribosomes, signal peptide binding proteins, SP receptor, Sec61, and Sec62 can also be identified in the Tgj1 PPI. Additionally, some dense granule proteins do not have any recognizable SP (e.g., TGME49_222170, Gra17) that can be post-translationally translocated to ER. Due to the important role of many secreted proteins in the pathogenicity of *T. gondii*, which involves various potential chemotherapeutic target organelles such as rhoptries, micronemes, dense granules, inner membrane complex, etc; the role of Tgj1 in vesicle secretion and transport seems very promising for future studies.

Gillies et al. [19] observe that Ydj1 shares a common genome-wide genetic interaction network with Hsp90, suggesting that it might be involved in the proper regulation of diverse Hsp90 clients rather than interacting promiscuously. This is expected because Tgj1 and other Hsp40s are at the first step of the Hsp70/Hsp90 cycle, recognizing different client proteins [32]. Similarly, *T. gondii* Hsp90 [33] and Tgj1 PPIs share similar biological pathways and subcellular localization. However, the intersection of Tgj1 and Hsp90 interactors only retrieved 102 proteins, mainly interactors associated with processes involving translation, protein folding, tRNA aminoacylation/translation, and cell redox homeostasis. While these appear to be pathways enriched in the Tgj1-Hsp90 axis, Tgj1 has another 234 interactors of its own, suggesting that this chaperone may have specific functions unrelated to the Hsp70/Hsp90 axis. Some specific Tgj1 interactors were already discussed above (mitochondria, membrane transport, and translocation), but the most notorious for their relevance to *T. gondii*, would be those associated with invasion/pathogenesis, cell shape, cell motility, and energy generation. *T. gondii* has a highly specialized machinery for cell motility/host cell invasion, and pathogenesis [48,49,50], and also shows significant flexibility in its metabolism and energy generation [51,52]. Studies on these subjects have emerged in recent years as they allow us to understand the parasite’s high capacity for adaptability to different environmental conditions. Beyond the focus of this study, the analysis also shows that the role of Hsp90 not associated with the Tgj1-Hsp90 axis is enriched in protein folding, RNA processing, transcription, and cell signaling, the latter could in turn share biological pathways. All this highlights the importance of *T. gondii* chaperones in various biological processes of the parasite, with each chaperone or complex revealing specialized roles that must be analyzed separately.

## 5. Conclusions

We identified cytosolic Tgj1 as an Hsp40 Ydj1/Hdj2 ortholog, which is involved in various biological processes in *T. gondii* as deduced from its interactome. The Tgj1 roles are mainly associated with translation, energy pathways, membrane proteins, vesicle transportation, and cell signaling. Although a pleiotropic role is expected for a chaperone, Tgj1 seems to play a central role in all these processes taking into account the negative phenotype score observed in tachyzoites. On one side, Tgj1 seems to interact on its own with proteins related to the transport of vesicles and membrane proteins, generation of energy, and invasion/pathogenesis. On the other hand, Tgj1 seems to contribute to the Hsp70/Hsp90 cycle, mainly through its interaction with proteins involved in processes associated with protein pathways (mainly translation and protein folding) and cell redox homeostasis. Interestingly, previous work showed that phosphorylation of the α-subunit of eukaryotic initiation factor 2 (eIF2α, TGME49_313230), an interactor of the Tgj1-Hsp90 axis, is associated with stress and bradyzoite differentiation in *T. gondii* [53]. More recently, it was also observed that *T. gondii* eIF2α and eIF2α kinase also respond to oxidative stress [54]. In the future, it would be interesting to establish whether the Hsp70/Hsp90 cycle initiated by Tgj1 has any implication in the response associated with this regulation of translation by phosphorylation of eIF2α. It is also interesting to study the PPIs of each chaperone and the Tgj1-Hsp90 axis separately since it shows specializations for each of them. In the case of specific pathways of Tgj1, its role in energy metabolism, invasion, motility, and pathogenesis is interesting for future studies, all being processes related and relevant for the adaptation of *T. gondii* to different environmental contexts.

## 6. Limitations

Co-IP assays have allowed the identification of a wide variety of *T. gondii* proteins from different metabolic pathways and diverse functions. The results consistently associate Tgj1 with various biological pathways. However, the direct interaction of each of the identified proteins must be confirmed by further experiments. Likewise, many interactors could not be detected, either because of their transient interactions, the conditions of the analysis, the presence in small quantities, or the requirement of specific extraction methods. For example, the presence of Tgj1 in the nucleus is lower than in the cytoplasm. The extraction of nuclei and a subsequent co-IP could enrich nuclear interactors. Finally, the analysis in this work involves the functions of Tgj1 in the in vitro lytic cycle of *T. gondii* under normal growth conditions. In the future, the interactome should be analyzed under different stress conditions. 

## Figures and Tables

**Figure 1 proteomes-11-00009-f001:**
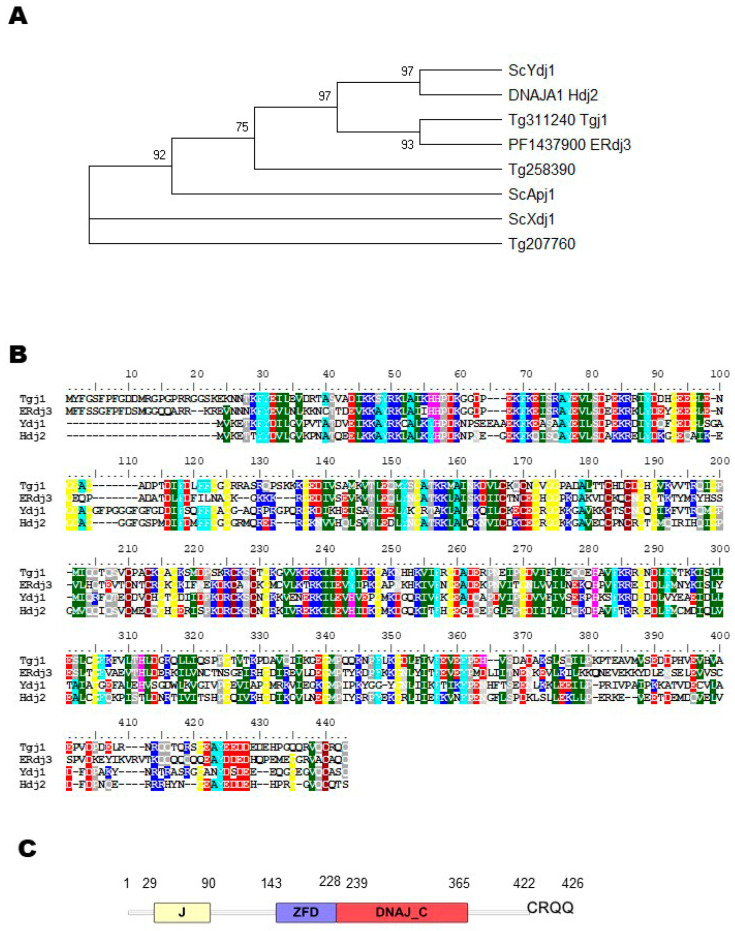
Evolutionary analysis by maximum likelihood method and sequence alignment. (**A**) Condensed Neighbor-joining tree of amino acid sequences. The tree is drawn to scale, with branch lengths measured in the number of substitutions per site. The analysis was performed in MEGA X, by evolutionary analysis using the UPGMA method with a Boostrap value of 1000. The sequences used were *T. gondii* Hsp40 type I (TGGT1_311240: Tg311240 Tgj1), putative DnaJ protein (TGGT1_258390: Tg258390), and a protein that contains a DnaJ domain in the C-terminal region (TGGT1_207760: Tg207760); *P. falciparum* PF3D7_1437900 (ERdj3); yeast Ydj1 (UniProt A6ZS16), Apj1 (UniProt P53940), and Xdj1 (UniProt P39102); and Human Hdj2 (UniProt P31689). (**B**) Tgj1, Erdj3, Ydj1, and Hdj2 amino acid sequences were aligned by Clustal W. (**C**) Protein domain architecture of Tgj1.

**Figure 2 proteomes-11-00009-f002:**
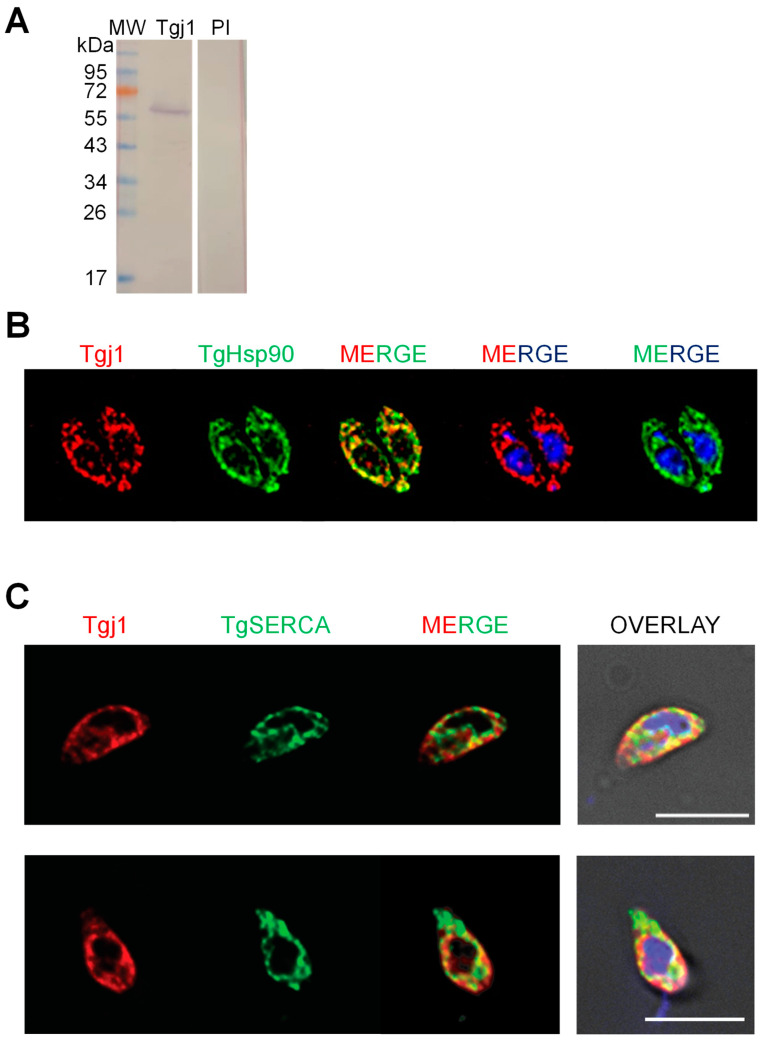
Analysis of native Tgj1. Subcellular localization was analyzed by indirect immunofluorescence of intracellular tachyzoites: (**A**) Western blot of *T. gondii* tachyzoite lysates with mouse anti-Tgj1 or Preimmune (PI) antibodies, both diluted 1:500. (**B**) Subcellular localization was analyzed in intracellular tachyzoites. Mouse anti-Tgj1 and rabbit anti-Hsp90 (green) antibodies. Antibodies were used at 1:200 dilution. (**C**) Subcellular localization was analyzed in extracellular tachyzoites. Mouse anti-Tgj1 (red) and guinea pig anti-TgSERCA (green) antibodies. Antibodies were used at 1:500 dilution. Nuclear DNA was labeled with DAPI (blue). The scale bars correspond to 5 μm.

**Figure 3 proteomes-11-00009-f003:**
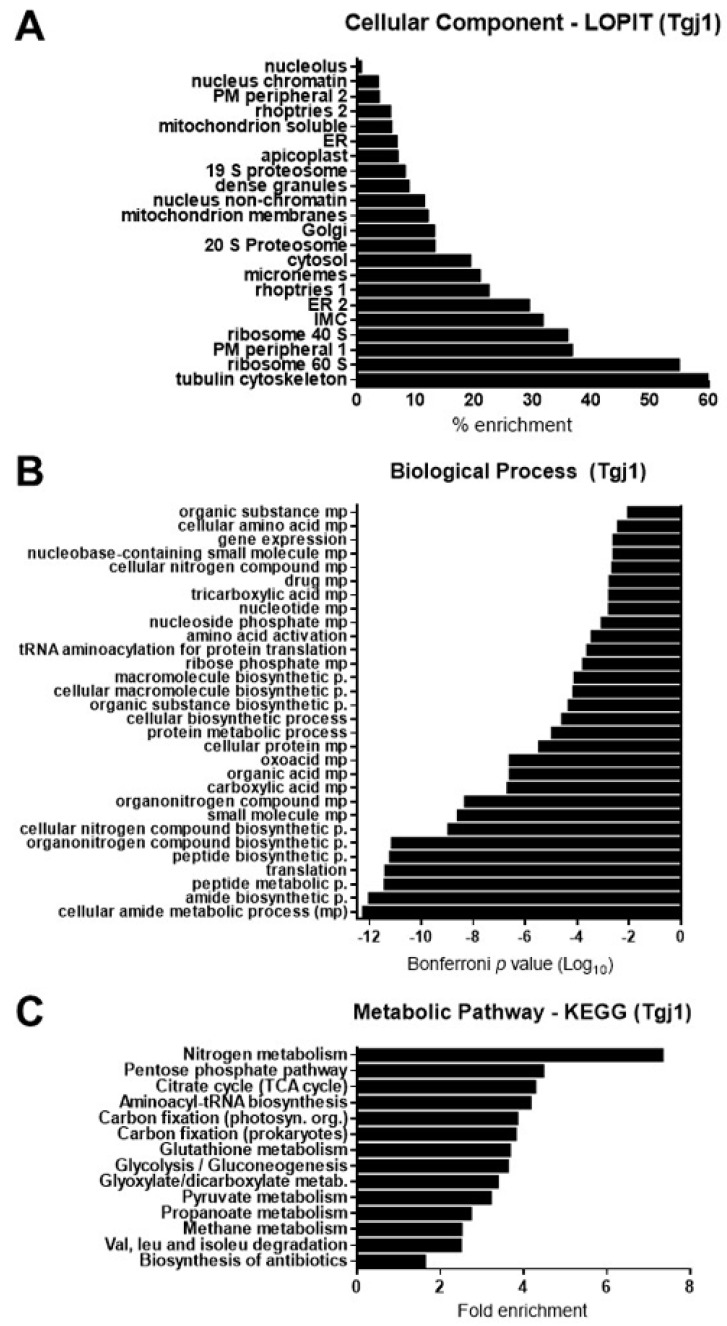
In silico analyses of putative Tgj1 interacting proteins. The Tgj1 interactor proteins were pulled down from tachyzoite (RH strain) extracts by immunoprecipitation with anti-Tgj1. These were subjected to SDS-PAGE and the bands were analyzed by mass spectrometry for their identification in the ToxoDB. Putative Tgj1 interactors are listed in Appendix A. (**A**) GO Cellular components and (**B**) Biological processes analysis. Bars represent genes associated with a specific GO term according to the Bonferroni *p*-value, in which the lowest value is the most significant. (**C**) KEGG annotation of Tgj1 interactors (*n* = 336). The fold enrichment of each pathway is shown on the x-axis.

**Figure 4 proteomes-11-00009-f004:**
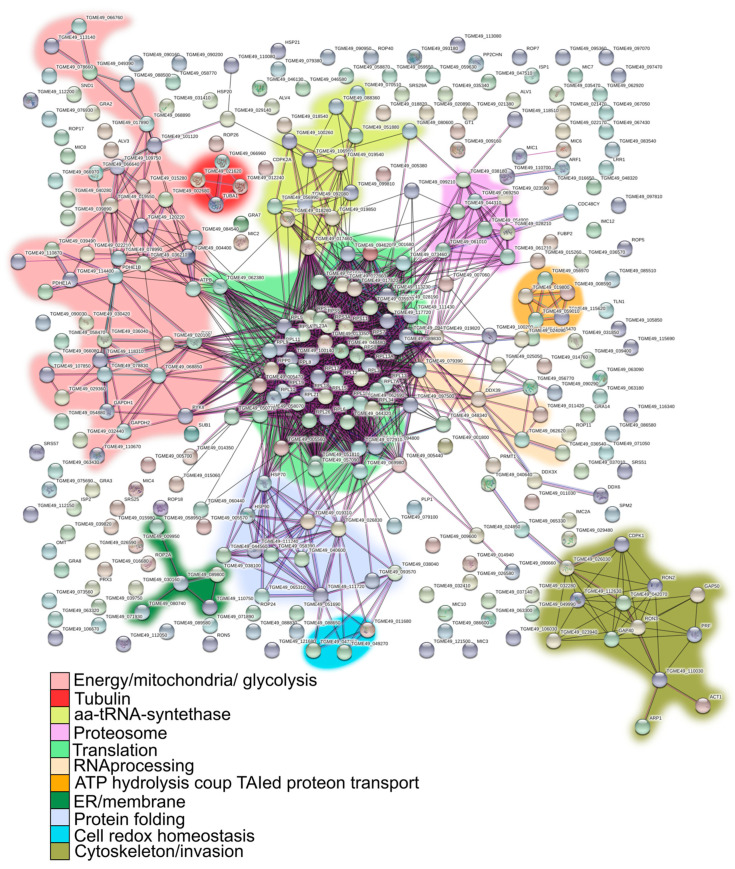
PPI interaction network and putative roles of Tgj1 in the biology of *T. gondii*. GeneIDs of Tgj1 interactors were analyzed by STRING software and their resulting networks were graphed. Green line: gene neighborhood, red line: gene fusions, blue line: gene co-occurrence. Statistic result = number of nodes: 174, number of edges: 111, average node degree: 1.28, avg. local clustering coefficient: 0.235, expected number of edges: 44, PPI enrichment *p*-value < 1.0 × 10^−16^.

**Figure 5 proteomes-11-00009-f005:**
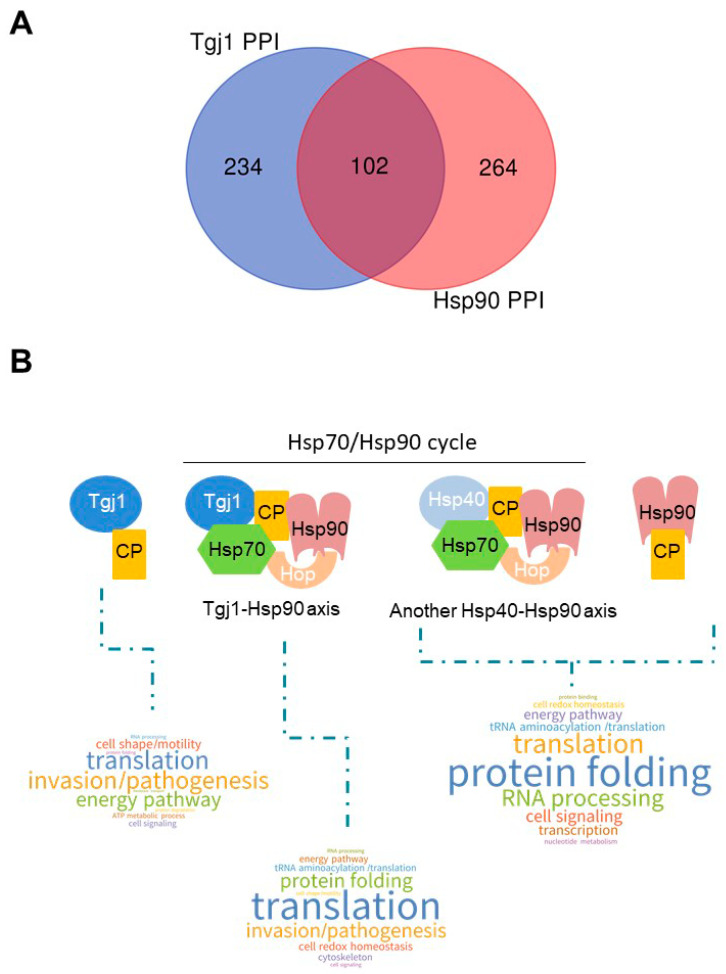
PPI chaperone network of Tgj1-Hsp90 axis. (**A**) Venn diagram of Tgj1 and TgHsp90 PPIs. (**B**) Word Cloud representation of enriched biological pathways associated with Tgj1 (subtracting Tgj1-Hsp90 PPI), Tgj1-Hsp90 axis, and Hsp90 (subtracting Tgj1-Hsp90 PPI) interactors (see Appendix A), excluding those whose biological process could not be inferred (ND in Appendix A). Biological pathways were defined based on subcellular location inferred by protein description, LOPIT and GO Biological Processes. In the case of Hsp90, interactors not related to the Tgj1-Hsp90 axis are inferred, which could be associated with the Hsp70/Hsp90 cycle initiated by another Hsp40 different from Tgj1, or by Hsp90 alone. ND, no data, was ruled out of the analysis. CP: client proteins. Observation: the Hsp70/Hsp90 cycle is dynamic. The figure shows only one moment of that cycle to graph what would be observed for each group of PPIs.

## Data Availability

All the raw data generated are available upon reasonable request to the corresponding authors.

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
