# Peer review of "Analysis of the Interactome of the Toxoplasma gondii Tgj1 HSP40 Chaperone"

_proteomes, 2023, doi:10.3390/proteomes11010009_

Round 1

Reviewer 1 Report (Previous Reviewer 1)

The manuscript has been revised as suggested.  I have no further comment.

Author Response

thanks

Reviewer 2 Report (Previous Reviewer 2)

The authors have thoroughly revised the manuscript and I think it should be published. 

Author Response

Thanks

This manuscript is a resubmission of an earlier submission. The following is a list of the peer review reports and author responses from that submission.

Round 1

Reviewer 1 Report

In this manuscript, the authors described the characterization of a novel protein (Tgj1) in Toxoplasma gondii. This protein is mainly localized in the cytoplasm of the parasite and interacts with a variety of proteins. Though the manuscript is well-written, additional experiments and major revision are needed before it could be accepted for publication.

1.      The title should be more precise. At this stage, no specific role of this protein has been clarified.

2.      Section 2.4: The authors should provide ethical clearance for using laboratory animals.

3.      Section 2.7: Line 163, please provide the target names of the primary antibodies. Otherwise it is unclear to know which protein(s) they aimed to co-localize with.

4.      Line 185, “10.000g” may be “10,000g”.

5.      Section 2.9: This approach has been replaced by more sensitive methods, and the authors should analyzed the data based on several repetitions, not just one experiment.

6.      Line 281: Co-localization with HSP90 is not a direct evidence of chaperone protein, the subtitle should be revised.

7.      Line 288: Supportive data for the overlapping of Tgj1 with the ER is missing.

8.      Figure 2D: The MW of Tgj1 is over 55 kDa (see line 259) should be explained. Also, the experiment with anti-Farnesyl antibody should be redone. The resolution is not acceptable. Further, an irrelevant antibody control should be included.

9.      Section 3.3: It is preferred to perform some co-localization with other molecule-molecule interaction approaches to verify the proteins that co-precipitated.

Reviewer 2 Report

Figure 2D, with the inhibitor, your band of Tgj1 seems to be thicker. Would it be good to quantify the intensity of the band and tie that to the expression level of the protein. 

Reviewer 3 Report

The manuscript "Elucidating the role of Tgj1 HSP40 chaperone in Toxoplasma" by Munera López shows interesting results about the characterization of a new chaperone in T. gondii and its interactors. Overall, the manuscript is well-written, and the results are well-presented and discussed. 

Minor suggestions:

Title: The manuscript did not show exactly what is the role of Tgj1, thus I suggest modifying the title.

Abstract:

-Please modify the conclusion in the abstract, as it is not possible to affirm how Tgj1 impacts the role of its interactors.

Introduction:

-Lines 47-48: Can it infect almost any nucleated cell or any nucleated cell?

-Lines 55-56: ...intermediate hosts (likely all warm-blooded animals, including humans). 

-Lines 60-61: "This occurs at a rate of 5-7 hours per replication cycle [5]." This is only for highly virulent strains, such as RH. 

- Line 75: Among the Hsps, the Hsp40 family (also called J-proteins) is classified into four  types according to their domain organization

- Lines 89-90: "The subcellular localization of Tgj1 was analyzed with an anti-Tgj1 antibody against the recombinant protein." Produced in mice using the recombinant protein?

Material and Methods

-Line 149: Please, specify the antibodies

- Line 186: What is  (25 l)?

Results

-3.2

I understand the concern of the authors regarding FTase II inhibitor against host cells or even parasite proliferation, but why not use it in intracellular parasites for investigating the differences in the Western blot profile? Maybe could use 100µM or even 200-400 µM for a short period.

Figure 2C: Maybe the authors should show the region (traced line) analyzed in the fluorescence image with Fiji.